# IgG4-Related Lymphadenopathy Mimicking Mediastinal Lymph Node Metastasis of Lung Cancer on ^18^F-FDG PET/CT

**DOI:** 10.3390/diagnostics15010041

**Published:** 2024-12-27

**Authors:** Ting-Chun Tseng, Hung-Pin Chan, Daniel Hueng-Yuan Shen, Chang-Chung Lin

**Affiliations:** 1Department of Nuclear Medicine, Kaohsiung Veterans General Hospital, Kaohsiung 813, Taiwan; tct.tseng@gmail.com (T.-C.T.); markscience05@hotmail.com (H.-P.C.); hyshen@vghks.gov.tw (D.H.-Y.S.); 2Department of Medical Education and Research, Kaohsiung Veterans General Hospital, Kaohsiung 813, Taiwan; 3Institute of Biomedical Sciences, College of Medicine, National Sun Yat-Sen University, Kaohsiung 804, Taiwan

**Keywords:** FDG, PET/CT, IgG4-related lymphadenopathy, lung cancer

## Abstract

We report a case of a 73-year-old man with minimally invasive lung adenocarcinoma, post-resection, evaluated with ^18^F-FDG PET/CT for suspected disease progression. Imaging showed increased FDG uptake in the right lower lung mass and systemic lymphadenopathy (mediastinal, supraclavicular, axillary, paraaortic, and iliac regions). The appearance of a stable lymph node and a clinical history of IgG4 lymphadenopathy suggested an inflammatory process, although malignancy in the lung mass and mediastinal nodes could not be excluded. Lobectomy confirmed the presence of lung adenocarcinoma, while radical lymph node dissection identified IgG4-related lymphadenopathy without metastasis. This case underscores the need for considering differential diagnosis of PET-positive lymphadenopathy, especially in patients with comorbid conditions that mimic or coexist with malignancy.

**Figure 1 diagnostics-15-00041-f001:**
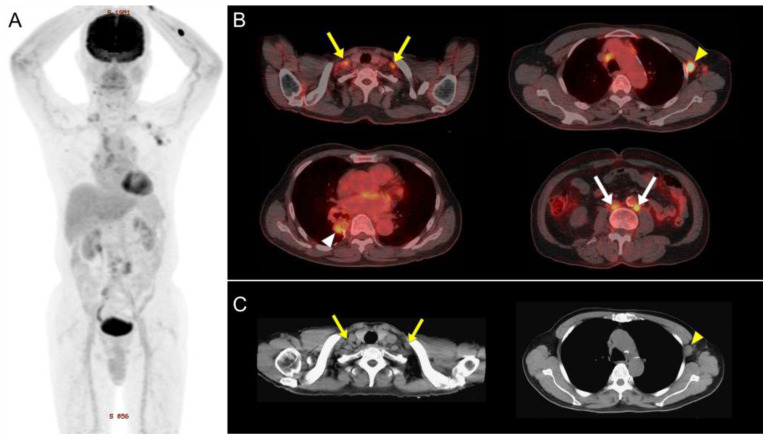
A 73-year-old male with stage IA minimally invasive lung adenocarcinoma, a status confirmed post-surgical wedge resection at age 66, underwent ^18^F-FDG PET/CT due to suspected disease progression on routine 3-month chest CT follow-ups. The maximum intensity projection coronal PET (**A**) revealed low-grade FDG activity in the right lower lung (RLL) tumor mass (SUVmax 4.6; (**B**), white arrowhead). Multiple FDG-avid lymphadenopathies were identified, including the mediastinal, bilateral supraclavicular (largest node measuring 12 mm; (**B**), yellow arrows), bilateral axillary (largest node 18 mm on the left; (**B**), yellow arrowhead), paraaortic (12 mm; (**B**), white arrows), and iliac nodes, with the left axillary lymph node showing the highest SUVmax of 7.7. The absence of systemic organ involvement made metastatic disease less probable. Serial chest CT scans, including one performed at age 67 (**C**), consistently showed no significant changes in the size or appearance of the bilateral supraclavicular ((**C**); yellow arrows) and axillary lymph nodes ((**C**); yellow arrowhead), reducing the likelihood of malignant spread. The symmetrical and systemic distribution of the lymphadenopathy suggested benign etiologies, such as inflammatory or lymphoproliferative disorders. Differentiation between lymphoma and inflammatory lymphadenopathy can be guided by two PET/CT observations and three clinical indicators: a high SUVmax liver ratio, elevated SUVmax in retroperitoneal lymph nodes, older age, a reduced erythrocyte sedimentation rate, and a low platelet count [1]. Given the low probability of lymphoma and lack of clinical correlation, the patient’s FDG-avid lymphadenopathies were deemed inflammatory. Notably, at the age of 59, the patient was diagnosed with IgG4 lymphadenopathy, confirmed by a left axillary lymph node biopsy revealing IgG4-positive transformed germinal centers and elevated serum IgG4 levels. This history supports IgG4-related lymphadenopathy as a probable diagnosis. Although rare, both IgG4-related inflammatory pseudotumors mimicking primary lung cancer [2,3] and lung cancer concomitant with IgG4-related disease [4] have been reported. For this patient, the RLL tumor mass and mediastinal lymphadenopathy required further evaluation to exclude cancerous processes. Differentiation using PET/CT can be challenging, as these pseudotumors often exhibit SUVmax values ranging from 3.4 to 10, with associated mediastinal and hilar lymphadenopathy showing SUVmax values between 2.8 and 3.9, overlapping with early-stage NSCLC (median SUVmax 3.2, range 0.7–13.3) [5]. Nevertheless, PET/CT remains valuable for identifying multi-organ involvement in IgG4-related disease prior to treatment [6] and monitoring the response to prednisone-based therapy [7], potentially reducing the need for biopsies. The patient subsequently underwent lobectomy, which revealed an acinar predominant adenocarcinoma, pT2aN0M0, stage IB. Radical lymph node dissection confirmed IgG4-related lymphadenopathy without metastasis. Histopathology demonstrated Bcl2 negativity in germinal centers, IgD positivity in mantle cells and the marginal zone, and an IgG4+/IgG+ plasma cell ratio of 45%, consistent with IgG4-related lymphadenopathy. This case underscores the importance of integrating clinical history, imaging comparisons, and alternative diagnoses, particularly when malignancy coexists with mimicking conditions [8]. Correlating PET/CT findings with clinical context is essential for accurate diagnosis and management.

## Data Availability

Not applicable.

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
