# Peer review of "IgG4-Related Lymphadenopathy Mimicking Mediastinal Lymph Node Metastasis of Lung Cancer on 18F-FDG PET/CT"

_diagnostics, 2024, doi:10.3390/diagnostics15010041_

Round 1

Reviewer 1 Report

Comments and Suggestions for Authors

This is an interesting case report of non cancerous lymphadenopathy following lobectomy for lung cancer on PET-CT.

There are a few comments:

1. Biopsy of the left axillary was performed before the surgical resection?

2. Was the case discussed in the lung cancer MDT to exclude any other malignant process?

3.  I would recommend to provide the staging PET-CT to compare the imaging findings

4. I also recommend to mention that this low volume lympadenopathy is quite symmetrical making inflammatory process and lymphoma the most common etiologies. 

5. References 2, 3 and 4 are not relevant to the case report as the primary lung lesion was cancerous and not a pseudotumour.

Author Response

This is an interesting case report of non cancerous lymphadenopathy following lobectomy for lung cancer on PET-CT.

There are a few comments:

  1. Biopsy of the left axillary was performed before the surgical resection?

Thank you for your valuable comments. We agree that additional clarification would improve the manuscript’s readability and comprehension. The biopsy of the left axillary lymph node was performed at age 59, seven years prior to the first surgical wedge resection at age 66, and 14 years prior to the lobectomy and radical lymph node dissection at age 73. This information has been revised in the manuscript.

  1. Was the case discussed in the lung cancer MDT to exclude any other malignant process?

Thank you for highlighting this point. Yes, a multidisciplinary team (MDT) meeting was conducted in 2024, involving radiologists, nuclear medicine specialists, and chest medicine experts to confirm the diagnosis and establish the treatment plan. The pathological report from the wedge resection in 2024 indicated an R1 resection, and the patient is currently undergoing adjuvant chemotherapy and radiotherapy. However, this information was not included in the article as the primary focus is on the diagnostic process in differentiating the cause of lymphadenopathy rather than the management of the primary malignancy. Thank you for your insightful comments.

  1.  I would recommend to provide the staging PET-CT to compare the imaging findings

T2a N0 M0 stage IB

  1. I also recommend to mention that this low volume lymphadenopathy is quite symmetrical making inflammatory process and lymphoma the most common etiologies. 

We appreciate this suggestion and agree that emphasizing the symmetrical distribution of lymphadenopathy would improve clarity. The wording has been revised accordingly for better clarification.

  1. References 2, 3 and 4 are not relevant to the case report as the primary lung lesion was cancerous and not a pseudotumour.

Thank you for pointing this out. We appreciate your suggestion to revise the references for better alignment with the case. The manuscript now includes relevant references discussing IgG4-related inflammatory pseudotumors mimicking primary lung cancer and cases of lung cancer coexisting with IgG4-related disease. This revision has been made.

Reviewer 2 Report

Comments and Suggestions for Authors

Dear Authors,

I have received an "interesting image" manuscript regarding IgG-4-related lymphadenopathy. I have several inputs:

1) Please include the size of the adenopathy

2) Please also include the relative interval of one CT scan to the other

3) Also include the histopathology figures in the manuscript

Comments on the Quality of English Language

-

Author Response

Dear Authors,

I have received an "interesting image" manuscript regarding IgG-4-related lymphadenopathy. I have several inputs:

1) Please include the size of the adenopathy

Thank you for the suggestion. We have now included the sizes of the lymphadenopathy in the revised manuscript. For example, the largest supraclavicular node measured 12 mm, the largest axillary node 18 mm, and the paraaortic node 12 mm. These details have been added.

2) Please also include the relative interval of one CT scan to the other

We appreciate this valuable input. The intervals between serial CT scans have now been specified in the manuscript, providing a clear timeline for the progression and stability of the lymphadenopathy.

3) Also include the histopathology figures in the manuscript

Unfortunately, the histopathology figures are not available for inclusion in this manuscript. However, we have provided a detailed description of the histopathological findings in the text, including IgG4-positive transformed germinal centers and the IgG4+/IgG+ plasma cell ratio, to ensure comprehensive documentation of the case.

Round 2

Reviewer 2 Report

Comments and Suggestions for Authors

Thank you for the revision. I have no further comments.

Comments on the Quality of English Language

-